# Discussing spiritual health in primary care and the HOPE tool—A mixed methods survey of GP views

**Ishbel Orla Whitehead**  *, Carol Jagger, Barbara Hanratty

Faculty of Medical Sciences, Campus for Ageing and Vitality, Population Health Sciences Institute, Newcastle Upon Tyne, United Kingdom

* orla.whitehead@newcastle.ac.uk

## Abstract

### Background

In the UK, the General Medical Council (GMC) and Royal College of General Practitioners (RCGP) require doctors to consider spiritual health in their consultations. There are documented barriers to discussion of spiritual health, and suggested tools to help overcome them.

### Aim

To investigate how comfortable general practitioners (GPs) feel about discussing spiritual health in the consultation, and whether a structured tool (the HOPE tool) would be helpful.

### Design and setting

A mixed-methods online survey completed by GPs in England.

### Method

A mixed methods online survey of practicing GPs in England asked about current comfort with the topic of spiritual health and use of spiritual history-taking tools. The acceptability of the HOPE tool was investigated using patient vignettes drawn from clinical practice.

### Results

177 GPs responded. 88 (49.71%) reported that they were comfortable asking patients about spiritual health. GPs felt most comfortable raising the topic after a patient cue (mean difference between pre and post cue 26%). The HOPE tool was viewed as acceptable to use with patients by 65% of participants, although its limitations were acknowledged. Qualitative data showed concerns about regulator (the GMC) and peer disapproval were major barriers to discussions, especially in the case of discordance between patient and doctor background.

### Conclusion

Only half of GPs are comfortable discussing spiritual health. Dedicated training, using a structured approach, with regulatory approval, may help overcome barriers to GPs

**Data Availability Statement:** This is human research participant data. Participants were asked to consent to anonymised data "becoming part of a data set which can be accessed by other users running other research studies at Newcastle

University and in other organisations. These organisations may be universities, or NHS organisations. [This] information will only be used by organisations and researchers to conduct research." The authors are concerned that this doesn't include consent for public data sharing, only for further research in universities or NHS organisations. Data will be shared upon reasonable request to the authors. The sentence "Data Access: While participants were not consented to allow public sharing of this data, data is available upon reasonable request to the authors." has been added to the manuscript. Data is stored at Newcastle University, and can be found at https://doi.org/10.25405/data.ncl.20939248. Access to the data can be obtained by contacting the author, or data managers at Newcastle University, rdm@ncl.ac.uk.

**Funding:** OW was funded by a post-CCT GP Fellowship funded by Health Education North East and Durham Dales and Easington Clinical Commissioning Group, and funds from the National Institute for Health Research (NIHR) School for Primary Care Research. Grant reference HEE REF 0150/8116. BH was part funded by the North East and North Cumbria Applied Research Collaboration. The funders had no role in study design, data collection and analysis, decision to publish, or preparation of the manuscript. The views expressed are those of the author(s) and not necessarily those of DDES CCG, the NIHR or the Department of Health and Social Care.

**Competing interests:** The authors have declared that no competing interests exist.

discussing spiritual health. Further research into the benefits, and risks, of discussion of spiritual health in the GP consultation is recommended.

## Introduction

In the UK, a General Practitioner (GP) is expected to be able to take a spiritual history from a patient to meet the obligations of the medical regulator and the Royal College of General Practitioners (RCGP) [1, 2]. Whilst taking a sexual or psychiatric history is a routine component of doctors' work, discussion of spiritual health is less established in consultations. Doctors' discomfort with this topic may not be because they believe that discussion is unimportant or outwith their role [3, 4]. Barriers include physician self-awareness [5–7], discordance in culture and religion between doctors and patients [4, 7, 8], practitioner discomfort [7], peer disapproval [7], time pressure [9] and difficulty identifying patients with spiritual needs [4, 10]. Some feel that spiritual health is only appropriate in mental health or palliative consultations [5, 6]. Relying on gut feeling to identify when to address the topic risks GP biases, rather than patient need, affecting whether those needs are addressed [11]. A lack of formal training, and a perceived lack of skills, appear to be major barriers to spiritual history taking [3, 6, 10].

Busy GPs in the UK may benefit from a concise tool to help overcome barriers to discussing spiritual health. The HOPE tool meets these requirements [5], as it provides both a clear structure for novice or uncomfortable practitioners, as well as a flexible and open approach for more experienced practitioners. The initial question is an open, non-religious one, 'what gives you hope in difficult times?' [5]. The tool is designed to be used flexibly [5], allowing it to be a useful addition to a GP's consultation skills, rather than a box-ticking exercise.

The aim of this study was to investigate how comfortable GPs feel discussing spiritual health with their patients, and to assess the potential benefit of a structured tool (HOPE) to overcome barriers to the discussion of spiritual health within the consultation.

## Method

An online survey was distributed to qualified GPs in England, using onlinesurveys.ac.uk. The survey was sent to all 211 Clinical Commissioning Groups (CCGs) to be included in CCG newsletters, as well as professional online groups, and forwarded to practice managers and GP practices directly. Consent was taken onlinein writing, prior to the start of the online survey. Ethics approval was sought and obtained from Newcastle University on 27 February 2019.

Questions collected demographic data from the participants, including sex, ethnicity and religion, and occupational characteristics.

Participants were asked to rate the following statements:

- I feel comfortable asking patients about their spiritual health

- I feel comfortable discussing spiritual health with patients at the end of life

- I feel comfortable discussing spiritual health with patients with poor mental health

A five-option Likert scale was used. Participants were asked which, if any, spiritual history taking tools they were aware of and use.

The HOPE tool was explained (see Table 1) and participants were asked whether they would feel comfortable using the HOPE tool, either being asked as a patient, or asking patients the questions.

**Table 1. The HOPE tool(5).**

There are a few structures or tools suggested to help GPs ask patients about their spiritual health.
This survey is about the HOPE tool, developed in the USA, to aid family physicians in taking a spiritual history.
The tool provides a series of prompts, and acts as a mnemonic.
**HOPE** stands for:
H- **Hope**- asking patients what gives them hope/sustains them
O- **Organised religion**- discussing whether patients interact with any form of organised religion
P- **Personal spiritual practice**
E- **Effects on care**- anything the patient needs you to know about how their spirituality impacts on their care, for example at the end of life, or refusal of certain treatments.

| | |
|---|---|
| **H**ope:<br>We have been discussing your support systems. I was wondering what is there in your life that gives you internal support? What are your sources of hope, strength, comfort and peace?<br>What do you hold on to during difficult times?<br>What sustains you and keeps you going?<br>For some people, their religious or spiritual beliefs act as a source of comfort and strength in dealing with life's ups and downs; is this true for you? | **O**rganised religion:<br>Do you consider yourself part of an organized religion?<br>How important is this to you?<br>What aspects of your religion are helpful and not so helpful to you?<br>Are you part of a religious or spiritual community? Does it help you? How? |
| **P**ersonal spirituality and **p**ractices:<br>Do you have personal spiritual beliefs that are independent of organized religion? What are they?<br>Do you believe in God? What kind of relationship do you have with God?<br>What aspects of your spirituality or spiritual practices do you find most helpful to you personally? | on medical care and **e**nd of life issues:<br>Has being sick (or your current situation) affected your ability to do the things that usually help you spiritually? (Or affected your relationship with God?)<br>As a doctor, is there anything that I can do to help you access the resources that usually help you?<br>Are you worried about any conflicts between your beliefs and your medical situation/care/decisions?<br>Would it be helpful for you to speak to a clinical chaplain/community spiritual leader?<br>Are there any specific practices or restrictions I should know about in providing your medical care? (e.g., dietary restrictions, use of blood products)<br>*If the patient is dying*: How do your beliefs affect the kind of medical care you would like me to provide over the next few days/weeks/months? |

Five vignettes (Table 2) were developed from a range of real clinical cases to reflect the socio-cultural diversity of the UK, as well as cover scenarios where all parts of the HOPE tool could be useful. They were presented in two parts: the first giving a scenario from clinical practice, and the second expanding that scenario with a patient cue that spiritual health may be relevant.

The participant was asked to rate the following statements for each vignette:

1. I would feel comfortable asking this patient about their spiritual health

2. I think the HOPE tool would be useful with this patient

**Table 2. Patient vignettes.**

| Patient name | Age | Ethnicity | Religious or similar background | Clinical issue | Intended spiritual component to consultation |
|---|---|---|---|---|---|
| Fatima | 32 | Arabic name, but used widely | Muslim | Post-natal depression | Isolation, spiritual crisis, mental illness (H, O, P, maybe E) |
| Derek | 80 | Suggest white British | Methodist | Oesophageal cancer and palliative care | End of life (E, maybe O, H maybe P) |
| Michael | 52 | Suggest white western European, likely British | Former Jehovah's Witness | Erectile dysfunction | Psychosexual/functional symptoms, possible spiritual crisis, potential change to consent for blood (O, P, E) |
| Olive | 72 | Unspecified, based on a patient from Europe | Likely Anglican or other mainstream Christian | Loneliness/frailty | Isolation, mild mental illness symptoms, possible functional symptoms.(H, O, P) |
| David | 24 | Likely British, ethnic background left open to the reader | Vegan/humanist | Acne, depression | Mental illness, compliance with meds (E) |

3. I would feel comfortable using the HOPE tool with this patient

A four-option Likert scale was used, with sections for free text comments.

## Data analysis

**Quantitative.**   Data analysis was conducted using the Stata SE 17.0 package [12]. Associations between binary variables were assessed by McNemar's test. Data were aggregated where small numbers require it for statistical analysis.

**Qualitative.**   Qualitative data on barriers, facilitators, and use of the HOPE tool in discussion of spiritual health were analysed using a deductive thematic analysis, based upon a priori themes from the literature. A four step process was used: [13] immersion in the data, stratifying to identify themes by comparing and contrasting similar codes, review of categories, and finally drawing these together to identify the central themes. Outlying cases were examined, to identify insights from those most and least comfortable with the topic.

## Public and Patient Involvement (PPI)

Six members of VOICE, a network of public, patients and carers (https://www.voice-global.org/about/), joined a meeting to discuss the findings from the survey. They expressed mixed views about both the topic and the HOPE tool. Some felt HOPE is a respectful and innocuous way to structure a discussion on the topic; all participants asserted that holistic, humanitarian care was essential.

## Quantitative results

One hundred and seventy-seven GPs responded. The majority (63%) were women, of white British origin (79%), and 99% had trained as GPs in the UK (Table 3). Seventy per cent of respondents stated they had a religion, with 63% Christian, and 7% other religions.

**Comfort discussing spiritual health and the effect of cues.**   Half of respondents reported they felt comfortable discussing spiritual health in general. Comfort with the topic varied according to the vignette topic, with a statistically significant ($p<0.05$) effect of patient cue on response (Table 4). Respondents were most comfortable discussing spiritual health in relation to end of life care (mean agree/strongly agree = 81% of respondents), and least comfortable in the erectile dysfunction vignette (mean agree/strongly agree = 39% of respondents.) The effect of a cue was greatest in the erectile dysfunction vignette (98% being more comfortable discussing spiritual health post-cue than pre-cue), and least in the end of life vignette (31% being more comfortable discussing spiritual health post-cue than pre-cue).

## The HOPE tool

**Use of history taking tools and comfort with the HOPE tool.**   The majority (94%) of respondents stated they never use a tool to support discussion of spiritual health. Most (77%) would be comfortable being asked the questions in the HOPE tool as a patient, and 65% would feel comfortable using the HOPE tool with their patients. While most respondents who felt comfortable using the HOPE tool were already comfortable discussing the topic, 16% of respondents uncomfortable with the topic felt they would be comfortable using the HOPE tool.

**Does concordance or discordance between doctor and patient have an effect on comfort?.**   Concordance between participant identifying as ethnic majority or minority and the vignette being likely ethnic majority or minority gave a significant difference in comfort with the topic. (estimated difference 0.2775, 95%CI (0.1961, 0.3589), McNemar's test). There was

**Table 3. Characteristics of respondents.**

| | Number of Participants (n = 177) | % |
|---|---|---|
| **Sex** | | |
| Male | 65 | 37 |
| Female | 111 | 63% |
| **Ethnic Group** | | |
| White British | 139 | 79% |
| Any other White background, | 7 | 4% |
| White Irish | 5 | 3% |
| Indian | 6 | 3% |
| Any other Mixed / Multiple ethnic background | 4 | 2% |
| Other background | 15 | 9% |
| **Religion** | | |
| Christian | 110 | 63% |
| Other | 12 | 7% |
| None | 49 | 28% |
| **Country of primary medical qualification** | | |
| England | 144 | 81% |
| Scotland | 14 | 8% |
| Elsewhere in Europe | 8 | 5% |
| Asia | 6 | 3% |
| Africa or Americas | 5 | 3% |
| **Country of GP training** | | |
| England | 168 | 95% |
| Scotland | 5 | 3% |
| Other | 4 | 2% |

no evidence of a significant difference of comfort with discussing spiritual health with concordance of faith (estimated difference of 0.0880 95% CI (-0.0034, 0.1793), McNemar's test).

## Qualitative results and analysis

**The HOPE tool.** Views on the HOPE tool were mixed. Some, especially those not comfortable discussing spiritual health, were positive: "I think this would be incredibly useful," and "I agree the HOPE tool would be very useful, but would need to practice using it before I feel completely comfortable." The starting HOPE question was praised as "a rather wonderful

**Table 4. Comfort with discussing spiritual health pre and post cue, compared with comfort using the HOPE tool.**

| Patient name | Number of participants agreeing they are comfortable discussing spiritual health | | Number of participants agreeing they are comfortable using the HOPE tool | | Number of participants uncomfortable discussing spiritual health who would be comfortable using the HOPE tool | |
|---|---|---|---|---|---|---|
| | **Pre-cue** | **Post-cue** | **Pre-cue** | **Post-cue** | **Pre-cue** | **Post-Cue** |
| Fatima (post-natal depression) | 76 (43%) | 142 (80%) | 67 (38%) | 120 (68%) | 13 (7%) | 6 (3%) |
| Derek (Palliative care) | 136 (77%) | 152 (86%) | 121 (68%) | 137 (77%) | 7 (4%) | 3 (2%) |
| Michael (Erectile dysfunction) | 17 (10%) | 121 (68%) | 15 (8%) | 93 (53%) | 4 (2%) | 3 (2%) |
| Olive (Loneliness) | 124 (70%) | 151 (85%) | 106 (60%) | 130 (73%) | 8 (5%) | 1 (<1%) |
| David (Acne and depression) | 53 (39%) | 90 (51%) | 46 (26%) | 73 (41%) | 12 (7%) | 8 (5%) |

question. . ." However, others suggested that the 'hope' question could be inappropriate in a palliative context, or patronising.

People who were comfortable addressing the topic with their existing consultation skills felt that tools such as HOPE can be too constraining and disrupt the flow of a consultation. One participant explained "The HOPE tool seems useful in opening the conversation, but once patient has revealed their spiritual side, a conversation that is more free flowing that explores their view would be far more useful than a tool." The HOPE tool was criticised for its length.

**Discordance.** The challenges of discordance in culture and faith between doctor and patient were developed within the comments. Discordance caused discomfort: "Steer well clear. . . abusive cult", and "I do not know enough about Jehovah's Witness,[sic] and would not like to be negative." There was concern about causing inadvertent offence to the patient where the doctor and patient have discordant beliefs and background. One patient's status as a religious authority figure could be "intimidating". Concordance increased comfort: "Much easier with someone who shares the same faith as me", "I would be comfortable asking what he is reading in the bible and discussing passages that may bring comfort to him and his wife if they wanted" and "easier if I share his culture". Another respondent disclosed praying with a patient where faith was shared. Concordance or discordance of culture and ethnicity appears to affect comfort with the topic, with fears of regulator disapproval where there is discordance: "As a male white GP [I] would feel it was intrusive," and "I am a white male asking a brown female about her beliefs. Should I just refer myself to the GMC to save the patient the bother."

**Barriers to discussing spiritual health.** Discussion of the topic and the HOPE tool were felt to take time away from physical and mental issues, for example: "This is ridiculous . . . I have not got time for her spiritual health." Spiritual health is labelled "not a priority" for the busy GP. A participant felt spiritual health assessment would be incorporated into the consultation in an "ideal world", but "high pressure [and] increasing complexity. . . I must focus on the clinical issues". Respondents suggested discussion could be delegated to others in the team, or third sector resources, feeling spiritual health is "not my role".

The need for a patient-led cue was mentioned by many respondents, supporting the quantitative findings: "once the cue is there, I could lead on." "I would feel uncomfortable asking spiritual questions without the patient bringing it up first." One participant referred to these cues as 'faith flags'. Participants with faith had concerns about perceptions of proselytising, for example: "A perception of those believers that they will breach rules and be criticised stops many discussing spiritual health," and "As a practicing evangelical Christian I could be in a lot of trouble for 'imposing my belief system' on vulnerable patients."

Lack of training was a barrier: "I would like to discuss more spiritual issues . . . of course time and my skills may be lacking," and "I don't feel confident that I have the language/ phrases needed to discuss it." Some participants were resistant to accessing training: "Nor do I have any training in it- nor want it." While some participants stated the topic was not addressed in GP training, a GP trainer mentioned that they do raise the topic in training. A few participants mentioned that they had sought training on the topic via the Christian Medical Fellowship's Saline Solution course. Lack of training appears to be a source of discomfort with the topic.

Participants repeatedly mentioned concerns about the opinion of the UK regulator, the General Medical Council (GMC), for example, "After the way the GMC has pilloried doctors who have discussed faith. Are you mad. Why would you give the GMC yet another reason to go after you." One participant disclosed peer disapproval by senior colleagues, despite following GMC guidance on the topic.

## Discussion

### Summary of main findings

This is a large study incorporating qualitative and quantitative data on the topic of spiritual health in primary care, and the first to explore views of the HOPE tool in the UK. Key findings from this research are: the impact of patient cues on GP comfort with discussing spiritual health, the acceptability of the HOPE tool, barriers to discussing spiritual health. These are discussed in detail below.

The largest impact on comfort with discussing spiritual health appears to be the patient giving a cue that the topic may be relevant. While this is to be expected, a reliance on patients to raise the topic may mean inequity in addressing spiritual health needs [11]. The HOPE tool was perceived as useful and acceptable by most respondents, and therefore likely to be acceptable to GPs. The HOPE tool may offer a way into talking about spiritual health for people who are not happy with the topic.

Qualitative views on the use of the HOPE tool reflected the strengths and limitations of tools identified in previous research. Those who commented that they would use 'normal consulting skills' had rated themselves as already comfortable with the topic, with respondents disliking the idea of an 'extra piece of paper to fill in'. The main barriers to discussion of spiritual health mentioned were lack of time, discordance between doctor and patient beliefs, concerns about the regulator, and lack of training. Fear of referral the regulator (for example the GMC) appears to be a significant inhibiting factor for some respondents. Respondents were concerned that patient or peer perceptions of proselytizing could cause referral to the GMC.

### Comparison with other work

As far as we are aware, this is the first time the acceptability of the HOPE tool to practitioners has been formally assessed in this way. The need for a change to primary care training has been highlighted previously [14]. While training in the HOPE tool alone would not address the positivism and Cartesian dualism within medical training, it could give a ingress to the topic for those uncomfortable. Perceived cues from patients made a significant difference to doctors' responses, and their comfort in talking about spiritual health and using the HOPE tool. This reflects previous work that has emphasised the need for an open approach, responsive to cues [5]. Concerns about the use of tools have been found similarly in other studies, While tools should not be tick boxes [5, 6], many respondents preferred to rely on their own communication skills, as was also found in the literature [10, 15].

The FICA tool has been previously evaluated by GPs in Flanders, with similar reservations about its use found as given here for the HOPE tool, for example the restrictive and artificial nature of tools [6].

### Barriers to discussion

Concordance had been identified as an issue within the doctor patient relationship, and particularly discussion of spiritual health, and was explored in interviews with doctors and patients within the USA [8]. In common with our findings, the authors reported that concordance between doctors and patients could assist in discussing spiritual health. Lack of time was mentioned in the published literature [9, 16] and appeared in our qualitative data. The HOPE tool is designed as a tool and framework and should be used flexibly according to patient cues and the demands of the consultation. Training in the HOPE tool gives a structured and flexible framework to give confidence to address spiritual health, even where these barriers appear [5].

## Strengths and limitations

The survey attracted UK-wide responses from a population that is often difficult to recruit into research. Views expressed were varied and frank. The study was designed and conducted with patient and public participation, which should ensure that it remained patient focused. However, the respondents are a self-selected sample, and strong views may have prompted participation. The respondents were more likely to be female (63% vs 53%) [17] and less likely to be of black or minority ethnic origin (14% vs 25%) [17] than the wider UK GP population. While the majority of respondents were from the North East of England, the data did not show a difference in response by geographical area. Participants were more likely to have a primary medical qualification (PMQ) from the UK (89% vs 79%), less likely to be International Medical Graduates (6% vs 13%) and equally likely to have their PMQ from elsewhere in Europe (5% vs 5%) than the GP population. Most (98%) respondents completed their GP training in the UK. Non-Christian religions were combined due to small numbers, limiting analysis of the effect of religious affiliation. Analysis of concordance and discordance of background was limited by low numbers of respondents from UK minority ethnicities.

## Implications for research and practice

The GMC and the RCGP expect GPs to include spiritual health in care, however, half of respondents are uncomfortable with the topic. Respondents to the survey reported that they had not had any training in this area, and none of the RCGP e-learning CPD modules make any reference to 'spiritual'. This is an important omission. Training is offered by some religious medical organisations, which may result in bias. The GMC guidance on discussing spiritual matters with patients [18] does not appear to engender confidence in the topic, as respondents named fear of regulatory involvement as a barrier to discussion, especially in cases of discordance between doctor and patient in terms of ethnicity, age or religious background. This barrier to discussion is recognised in the literature [7]. Robust and clear training, with guidelines and a structure, e.g. the HOPE tool, could help overcome such concerns.

Further research into the effects of concordance and discordance of ethnicity and faith/religion between patients and doctors, and of how self-awareness of these factors affects our communication, is needed to explore this topic, with more diverse recruitment.

## Conclusions and recommendations

This study suggests that a structured approach to discussing spiritual health (as offered by the HOPE tool) would be acceptable and useful for GPs who are uncomfortable with the topic. However, to embed spiritual health in primary care consultations in the future, dedicated training is likely to be required. This study did not address what GPs should do with the information they gather. Health services are not best placed to provide spiritual care [15], and further investigation is needed into how to ensure those with spiritual needs are directed to appropriate services, (e.g. social prescribing or chaplaincy). Increasing comfort with the topic, e.g. training in the HOPE tool, could allow such referrals to occur, and better meet patients' needs.

## Supporting information

**S1 File. Supplementary file: Whitehead O, Jagger C, Hanratty B.** What do doctors understand by spiritual health? A survey of UK general practitioners. BMJ Open 2021;11:e045110. doi:10.1136/bmjopen-2020-045110.
(PDF)

## Acknowledgments

Thank you to all the GPs who participated, and all those in CCGs and CRNs who helped with recruitment. Thank you to JISC online surveys software. Thank you to Voice North members who helped shape the study and analysis.

## Author Contributions

**Conceptualization:** Ishbel Orla Whitehead, Carol Jagger, Barbara Hanratty.

**Data curation:** Ishbel Orla Whitehead.

**Formal analysis:** Ishbel Orla Whitehead.

**Funding acquisition:** Ishbel Orla Whitehead.

**Investigation:** Ishbel Orla Whitehead.

**Methodology:** Ishbel Orla Whitehead, Carol Jagger, Barbara Hanratty.

**Project administration:** Ishbel Orla Whitehead.

**Resources:** Ishbel Orla Whitehead.

**Supervision:** Carol Jagger, Barbara Hanratty.

**Validation:** Carol Jagger, Barbara Hanratty.

**Visualization:** Ishbel Orla Whitehead.

**Writing – original draft:** Ishbel Orla Whitehead.

**Writing – review & editing:** Ishbel Orla Whitehead, Carol Jagger, Barbara Hanratty.

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
