## [Decision Letter · Decision Letter 0]

28 Jun 2022

PONE-D-22-14015Discussing spiritual health in primary care and the HOPE tool- A mixed methods survey of GP viewsPLOS ONE

Dear Dr. Whitehead,

Thank you for submitting your manuscript to PLOS ONE. After careful consideration, we feel that it has merit but does not fully meet PLOS ONE’s publication criteria as it currently stands. Therefore, we invite you to submit a revised version of the manuscript that addresses the points raised during the review process.

We look forward to receiving your revised manuscript.

Kind regards,

Luigi Lavorgna

Academic Editor

PLOS ONE

Journal Requirements:

3. We noted in your submission details that a portion of your manuscript may have been presented or published elsewhere. [The paper: "What do doctors understand by spiritual health? A survey of UK general practitioners" is related to this submission, in that the data were gathered in the same survey, and therefore table 1 in both papers is the same. That paper analysed data on how the participants defined "spiritual health", whereas this paper analyses the data about use of the hope tool, and discussing spiritual health in the consultation. While there is minimal overlap, data from different questions within the survey are presented in the two different papers, and therefore this is not dual publication of the same data. ] 

Please clarify whether this publication was peer-reviewed and formally published. If this work was previously peer-reviewed and published, in the cover letter please provide the reason that this work does not constitute dual publication and should be included in the current manuscript.

6. Please ensure that you refer to Figure 1 in your text as, if accepted, production will need this reference to link the reader to the figure.

Reviewers' comments:

Reviewer's Responses to Questions

**Comments to the Author**

1. Is the manuscript technically sound, and do the data support the conclusions?

Reviewer #1: Partly

2. Has the statistical analysis been performed appropriately and rigorously? 

Reviewer #1: Yes

3. Have the authors made all data underlying the findings in their manuscript fully available?

Reviewer #1: Yes

4. Is the manuscript presented in an intelligible fashion and written in standard English?

Reviewer #1: Yes

5. Review Comments to the Author

Reviewer #1: Abstract (results): please correct the percentage of GPs responders in 49.71

Introduction: line 48. When you talking about the importance of spirituality in patients please check studies in neurology and in oncology that have proved the importance of spirituality in some disease ( Sparaco M, Miele G, Abbadessa G, Ippolito D, Trojsi F, Lavorgna L, Bonavita S. Correction to: Pain, quality of life, and religiosity in people with multiple sclerosis. Neurol Sci. 2021 Dec 10. doi: 10.1007/s10072-021-05814-x. Epub ahead of print. Erratum for: Neurol Sci. 2021 Nov 23;: PMID: 34890003.)

Discussion: in which other studies the structured tool (HOPE) was used ?

6. PLOS authors have the option to publish the peer review history of their article (what does this mean?). If published, this will include your full peer review and any attached files.

Reviewer #1: No

---

## [Author Response · Author response to Decision Letter 0]

5 Sep 2022

Thank you very much for taking the time to review our manuscript.

Reviewers' comments:

Reviewer's Responses to Questions

Comments to the Author

1. Is the manuscript technically sound, and do the data support the conclusions?

Reviewer #1: Partly

2. Has the statistical analysis been performed appropriately and rigorously? 

Reviewer #1: Yes

3. Have the authors made all data underlying the findings in their manuscript fully available?

Reviewer #1: Yes

4. Is the manuscript presented in an intelligible fashion and written in standard English?

Reviewer #1: Yes

5. Review Comments to the Author

Reviewer #1: Abstract (results): please correct the percentage of GPs responders in 49.71

This has been made more exact.

Introduction: line 48. When you talking about the importance of spirituality in patients please check studies in neurology and in oncology that have proved the importance of spirituality in some disease ( Sparaco M, Miele G, Abbadessa G, Ippolito D, Trojsi F, Lavorgna L, Bonavita S. Correction to: Pain, quality of life, and religiosity in people with multiple sclerosis. Neurol Sci. 2021 Dec 10. doi: 10.1007/s10072-021-05814-x. Epub ahead of print. Erratum for: Neurol Sci. 2021 Nov 23;: PMID: 34890003.)

Thank you for your interest in this topic. Spiritual health does appear to be an important aspect of patients’ health, in multifactorial ways. Much of the research in this area can be difficult to compare and analyse, as religiosity and spiritual health and similar concepts are often conflated. We have not sought to justify why spiritual health is an important aspect of health in this study- the Royal College and the regulator have stated that GPs in the UK should be able to address a patient’s spiritual health. We have also not sought to present a definition of spiritual health to participants, as participants were asked to define the term for themselves. (Presented in our other publication: Whitehead O, Jagger C, Hanratty B. What do doctors understand by spiritual health? A survey of UK general practitioners. BMJ Open 2021). This has allowed this survey to consider discussions about spiritual health in the broadest sense, including religiosity, spirituality, but encompassing whatever the term meant to the participant.

Discussion: in which other studies the structured tool (HOPE) was used ?

As far as we are aware, this is the first time the acceptability of the HOPE tool to practitioners has been formally assessed in this way. Editorials and teaching sessions have included the tool, often alongside FICA, and/or BELIEF. The benefits of the HOPE tool are that while it can be used for religious patients, there is no assumption of a faith practice, and is flexible to meet a variety of patient needs. “As far as we are aware, this is the first time the acceptability of the HOPE tool to practitioners has been formally assessed in this way.” Has been added to the discussion.

6. PLOS authors have the option to publish the peer review history of their article (what does this mean?). If published, this will include your full peer review and any attached files.

Do you want your identity to be public for this peer review? For information about this choice, including consent withdrawal, please see our Privacy Policy.

Reviewer #1: No

---

## [Editor Report · Decision Letter 1]

4 Oct 2022

Discussing spiritual health in primary care and the HOPE tool- A mixed methods survey of GP views

PONE-D-22-14015R1

We’re pleased to inform you that your manuscript has been judged scientifically suitable for publication and will be formally accepted for publication once it meets all outstanding technical requirements.

Kind regards,

Luigi Lavorgna

Academic Editor

PLOS ONE
---

## [Editor Report · Acceptance letter]

28 Oct 2022

PONE-D-22-14015R1 

Discussing spiritual health in primary care and the HOPE tool- A mixed methods survey of GP views 

Dear Dr. Whitehead:

I'm pleased to inform you that your manuscript has been deemed suitable for publication in PLOS ONE. Congratulations! Your manuscript is now with our production department. 

Kind regards, 

on behalf of

Dr. Luigi Lavorgna 

Academic Editor

PLOS ONE